# Numerical Analysis and Experimental Verification of Damage Identification Metrics for Smart Beam with MFC Elements to Support Structural Health Monitoring

**DOI:** 10.3390/s21206796

**Published:** 2021-10-13

**Authors:** Andrzej Koszewnik, Kacper Lesniewski, Vikram Pakrashi

**Affiliations:** 1Department of Robotics Control and Mechatronics, Faculty of Mechanical Engineering, Bialystok University of Technology, Wiejska 45C Street, 15-351 Bialystok, Poland; k.lesniewski@doktoranci.pb.edu.pl; 2UCD Centre for Mechanics, Dynamical Systems and Risk Laboratory, School of Mechanical and Materials Engineering, University College Dublin, D04 V1W8 Dublin, Ireland; vikram.pakrashi@ucd.ie

**Keywords:** Structural Health Monitoring, piezoelectric, energy harvesting, damage detection, macro fiber composites (MFC), damage sensitive feature, finite element method (FEM)

## Abstract

This paper investigates damage identification metrics and their performance using a cantilever beam with a piezoelectric harvester for Structural Health Monitoring. In order to do this, the vibrations of three different beam structures are monitored in a controlled manner via two piezoelectric energy harvesters (PEH) located in two different positions. One of the beams is an undamaged structure recognized as reference structure, while the other two are beam structures with simulated damage in form of drilling holes. Subsequently, five different damage identification metrics for detecting damage localization and extent are investigated in this paper. Overall, each computational model has been designed on the basis of the modified First Order Shear Theory (FOST), considering an MFC element consisting homogenized materials in the piezoelectric fiber layer. Frequency response functions are established and five damage metrics are assessed, three of which are relevant for damage localization and the other two for damage extent. Experiments carried out on the lab stand for damage structure with control damage by using a modal hammer allowed to verify numerical results and values of particular damage metrics. In the effect, it is expected that the proposed method will be relevant for a wide range of application sectors, as well as useful for the evolving composite industry.

## 1. Introduction

Modern mechanical and civil structures are becoming increasingly flexible and complex with time. Beams, pipes, and cables in disparate areas of engineering (e.g., aeronautical, bridge, petroleum, renewable energy) remain critical structural elements which continue to degrade and remain susceptible to both external excitation and internal disturbances, leading to increasing risk and maintenance costs [1,2,3,4,5]. Critical components in these complex structures are typically designed around limit state principles for a certain level of damage tolerance and their maintenance and inspection schedules may not necessarily provide an appropriate evolution of damage due to logistic and procedural aspects, epistemic and aleatory uncertainties around such processes, and due to human effects.

In this context, damage detection or assessment of the current condition of structure can be addressed effectively through Structural Health Monitoring (SHM) by reducing the number and frequency of inspections, uncertainties, and also provide some information about the capacity of a system, which is not possible from the more popular visual inspections, thereby also increasing the value of information from such systems and eventually create more resilient structures. A wide range of SHM systems are widely studied by both academia and the industry, including vibration, optical, thermal, or impedance-based methods [6,7,8]. In particular, the vibration method is intensively developed by many researchers due to the simplicity of implementation of chosen sensors on real structures and fast detection of damage in the structures with the use of real-time monitoring systems. The instance of these considerations are papers [9,10], where the behavior of the intact and damaged mechanical structures has been assessed based upon natural frequencies, mode shapes, frequency response functions, and also power spectral density or based on spatial wavelet analysis [11].

Many investigations in this field have shown that one-layer piezoelectric patches or piezo harvesters with lightweight fiber materials allow identifying changes in different kinds of vibrating structures [12,13,14,15,16]. The relevant properties of these element, especially their fragility or extreme value effects allow to use them in many applications to monitor structure or detect damage [17,18,19,20,21]. A core aspect behind energy harvesting not being a part of SHM is the relative lack of sector specific examples and benchmarks, where the damage detection markers are compared in terms of their performance. Such markers can relate to damage existence, location, extent, and a combination thereof and it is important to discover and distinguish which markers are relevant for which purposes. In recent times, there has been some effort in creating benchmarks yielding sector specific challenges and energy harvesting sensors, along with markers for damage detection. A recent work [22] uses energy harvesting (EH) systems to assess the leak localization in water pipes. Here, the authors investigated several pipes with different widths of leak to propose and calibrate a leak index based on the monitoring voltage from a piezoelectric energy harvester (PEH) and the power spectrum of the output signal generated from particular polyvinylidene fluoride (PVDF) piezoelectric transducers. Subsequently, the use of Pb-free biomolecular piezoelectrics was also used to enhance SHM of water pipes [23]. Similar examples are available for bridge monitoring under operational conditions via Pb-zirconate titanate (PZT) patches [24,25]. Similar investigations have been also performed with piezoelectric PVDF sensors for wireless monitoring of tension conditions in a cable stayed bridge [26]. Similar applications have also been considered for the aerospace sector in terms of component monitoring in airplanes [27], including those harvesting energy from fuselage vibration with one-layer piezo transducers [28].

While the potential of an energy harvesting system for supporting SHM is established in principle, the implementation of it in specific engineering sectors is still fraught with several questions around interpretation and performance of responses, their analyses, and metrics developed for monitoring features of interest. An initial computational model is helpful in this regard to assess sensor location, potential impact of nonlinearities of the energy harvesting element, and the eventual translation of such information into designing appropriate SHM systems. The advantage of an energy harvesting-based monitoring often lies in the low-power aspect of it and the ease of use, which can lead to the possibility of extensive instrumentation. Model updating and digital twinning are also becoming more common in such industries and, consequently, a work like this will also provide a connectivity of harvesters integrated to such updating processes, for operational structures and those which are evolving through varied technological readiness levels.

There exists papers which address elements of the abovementioned challenges. The influence of non-linear geometric responses of a piezoelectric composite plate considering von Karman large strain theories into the classic plate solution has been investigated by using a 3D element model [29], with results indicating that the problem cannot be omitted especially when correct prediction of the stress-strain over the PEH is analyzed. A piezoelectric element modeled as a shell element acting under d31 effect of the crystal [30] noted that the effect of non-linearity is small and can be neglected, especially when commercial piezoelectric patches are used. Other scientific works in this field focused on introducing piezoelectric coupling to the shell element [31,32] and the results indicate the influence of non-linearity for piezoelectric laminated shell is significant and should be further analyzed. As a result, this led to developing investigations and modeling the piezo structure as a higher order layer-wise plate finite element considering piezoelectric coupling [33]. In summary, despite many scientific contributions related to formulations of plates and shells for piezoelectric laminated elements, there is a gap in verification of numerical results considering the shell finite elements of the piezoelectric element. This paper addresses this knowledge gap by carrying out numerical analysis and subsequently validating them against experimental results. 

Studies similar to what has been presented in this paper are also of particular relevance for new sensors that are being developed from environmental perspectives to avoid Pb-based systems [23] or multi-functional materials [34]. As their material properties and uncertainties become lower, the possibility of their use in many sectors get higher and the current work can make them better adapted to the composites sector where energy harvesting based SHM still requires significant research. While the presence of damage is often easier to detect with a sensor, the localization of such damages often provides further challenges, especially when the damages are relatively closely spaced. The impact of closely spaced damages have been studied before [11,20], and while there is a natural understanding that indicators of damage at some point would present a coalesced effect of two damages after a certain proximity, there is a paucity of literature in investigating quantitatively what the effects are for patch based systems. Furthermore, the use of composite materials for this topic is important as a benchmark due to their extensive use in new sectors, especially in marine environments [21]. Over time, their degradation, especially in the durability aspects for saline and harsh marine environments [35], will be of particular relevance around this topic. There is thus a need for a detailed numerical and experimental investigation of a relatively generic example which can be used in the future for similar studies, but also as an evidence base for current performance of patch-based energy harvesting and SHM, future adaption to new sensors, and to new structural systems and environments. This will also lend eventually to estimates on their lifetime risk levels [36] and comparative performances [37] around such levels, especially when the exposure is of a stochastic nature. 

Therefore, this paper is focused on damage identification in thin structures by using piezo fiber composites. In contrast to [20,21,23], the piezo harvester with three dimensional material in the piezoelectric fiber layer is considered to enhance the harvesting effect, especially in resonance regions with the host structure. To establish this, three different structures have been investigated, where the first one is intact, while the other two are damaged. The damage is introduced by drilling holes in the area of the beam closer to the end of the piezo-composite located closer the fixed end of the beam. The numerical results obtained from the finite element (FE) models of both sensors and also the damage indexes were determined next based on the frequency response functions, which subsequently allows for damage localization assessment. An experimental test on a real structure was finally carried out to verify the numerical results. 

The paper is organized as follows: Section 2 describes the methodology of modeling a piezo-harvester based on the First Order Shear Theory (FOST) and also presents the procedure of determining the shell finite element of the laminated structure of the piezo-composite. Section 3 presents the computational models of all structures (intact and damage) with a homogenous model of macro-fiber composite (MFC), which is also a core novelty of this manuscript. In Section 4, experimental investigations were carried out for all structures with the use of an impact modal hammer. The measurement signals from both MFC elements allow calculation of real values of damage matrices and for comparison with numerical results. Additionally, the approach allows assessing the damage location. Section 5 concludes the main findings of this work.

## 2. Mathematical Model of the Smart Structure with a Laminated Model of the MFC Element

The finite element model designed on the basis of the First Order Shear Theory (FOST) is mainly used in many applications consisting of thin structures like beams or plates to monitor their state via piezoelectric patches like macro-fiber composites [38]. The chosen finite element (FE) model is described by the laminated shell quadratic finite element with eight nodes: five mechanical degrees of freedom (three translations, two rotations) and three electrical DOFs related to the number of the piezoelectric-active layers of MFCs. In addition, this FE model can act as an equivalent single layer model to describe the mechanical behavior and as a layer-wise model to describe the electrical behavior. As a result, all the aforementioned performances of the proposed FE element caused that this element can simulate not only layered materials (e.g., laminated structural composites) with the piezoelectric layers embedded in the structure, but also non-layered materials (e.g., aluminum) with piezoelectric sensors attached to this structure. Another important aspect to note is that only displacements, forces applied to the structure (impulse inputs), and electrical potentials can be enough to obtain the frequency response functions of this structure and calculate the damage metrics properly. This makes the proposed FE model more useful to increase the computation efficiency, especially for real time SHM, without compromising damage identification accuracy.

The shell finite element, capable of simulating a plane, can be implemented in Ansys and solved using the curvature formulation, which must be at least quadratic to describe single or double curved shells with adequate accuracy [39]. Under such circumstances, the problem of interpolation of the curvature field should be considered in the same way as in the case of the issue of variables. In order to do this, it can be assumed that direction 1 is taken to be aligned to the fibers of a given lamina, direction 3 is aligned to the normal laminate direction, while direction 2 is obtained based on the right-hand rule. Indexes of all variables described in this manuscript are then given in the range of *i* = 1..3, *j* = 1..3, *k* = 1..3, *l* = 1..3. 

### Constitutive Equations and Electrical Assumptions of the Piezo-Electric Lamina and Its Finite Element Model

The piezoelectric MFC, as shown in Figure 1, is a five-layer smart composition with a single active layer of the thickness of *t_MFCa_*, two electrode layers of the thickness of *t_MFCe,_* and two Kapton layers of the thickness of *t_MFCk_*. As a result, this smart MFC element, attached to the host structures, is described in the form of constitutive equations given by Equation (1).
(1)$$\begin{array}{l} \sigma_{ij}=C_{ijkl}^{E}\varepsilon_{kl}-e_{kij}E_{k},\\ D_{i}=e_{ijk}\varepsilon_{kl}+d_{ik}^{\varepsilon }E_{k} \end{array}$$
where:

$\sigma_{ij}$—the laminate stresses; 

$D_{k}$—the electrical displacement; 

$E_{k}$—the electrical field; 

$C_{ijkl}^{E}$—the short-circuit elastic properties of the piezo-laminate;

$\varepsilon_{kl}$—the laminate strains; 

$e_{ijk}$—the piezoelectric coupling coefficients;

$d_{ik}^{\varepsilon }$—the dielectric properties of the piezo-laminate. 

**Figure 1 sensors-21-06796-f001:**
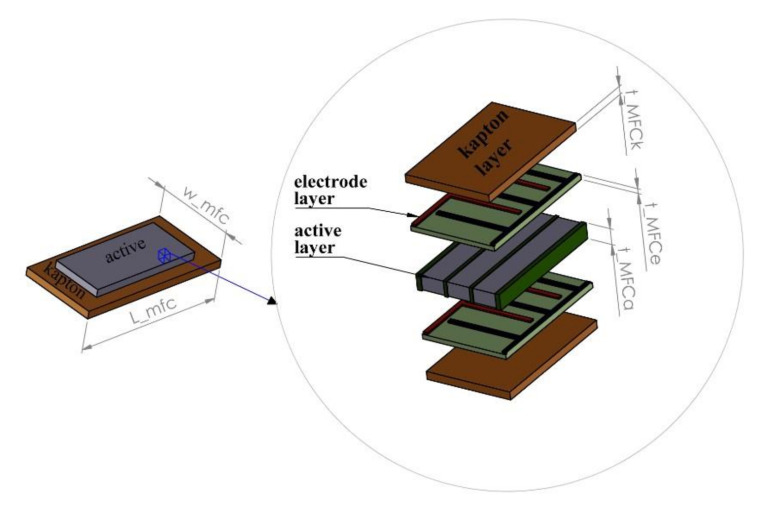
The structure of the macro-fiber composite.

From the measurement point of view, a proper polarization direction of the active element attached to the structure is required. In many applications, especially in those related to SHM, this problem has been solved by choosing the transverse direction of the active element polarization. As a result, Equation (1) can be simplified to the form given by Equation (2) when the Voigt notation is used, and also the normal transversal stress can be considered by omitting ($\sigma_{33}\approx 0$).
(2)$$\left\{\begin{array}{c} \sigma_{11}\\ \sigma_{22}\\ \sigma_{12}\\ \tau_{23}\\ \tau_{13}\\ D_{3} \end{array}\right\}=\left[\begin{array}{cccccc} Q_{11} & Q_{12} & 0 & 0 & 0 & -e{’}_{31}\\ Q_{12} & Q_{22} & 0 & 0 & 0 & -e{’}_{31}\\ 0 & 0 & Q_{66} & 0 & 0 & 0\\ 0 & 0 & 0 & Q_{44} & 0 & 0\\ 0 & 0 & 0 & 0 & Q_{55} & 0\\ e{’}_{31} & e{’}_{31} & 0 & 0 & 0 & d{’}_{33} \end{array}\right]\left\{\begin{array}{c} \varepsilon_{11}\\ \varepsilon_{22}\\ \varepsilon_{12}\\ \gamma_{23}\\ \gamma_{13}\\ E_{3} \end{array}\right\},$$
where the material matrix components are
$$\begin{array}{l} Q_{11}=C_{11}^{E}-\frac{C_{13}^{E2}}{C_{33}^{E}},\;Q_{12}=C_{12}^{E}-\frac{C_{13}^{E2}}{C_{33}^{E}},\;Q_{22}=C_{22}^{E}-\frac{C_{13}^{E2}}{C_{33}^{E}},\;Q_{66}=2\left(C_{11}^{E}-C_{22}^{E}\right),\;\\ Q_{44}=C_{44}^{E}+\frac{e_{15}^{2}}{d_{11}^{}},Q_{55}=C_{55}^{E}+\frac{e_{15}^{2}}{d_{11}^{}},\;e{’}_{31}=e_{31}^{}-\frac{C_{13}^{E}e_{33}^{}}{C_{33}^{E}},\;d{’}_{33}=d_{33}^{\varepsilon }-\frac{e_{33}^{2}}{C_{33}^{E}}. \end{array}$$

Application of smart elements in the form of macro fiber composites in mechanical structures requires also considering their mechanical and electrical properties. In the case of the mechanical behavior of the laminated structures, the Equivalent Single-Layer (ESL) approach has been used. Then, according to Figure 1, actuating and generalized forces and moments, like: shearing forces (Q), bending moments (M), normal moments (N), and torsion moments (T) in relation to the thickness of the piezo, can be written in the local coordinate system in the following forms: (3)$$\left\{\begin{array}{c} N_{x}\\ N_{y}\\ N_{xy} \end{array}\right\}=\int_{-t_{MFCa}/2}^{t_{MFCa}/2}\left\{\begin{array}{c} \sigma_{xx}\\ \sigma_{yy}\\ \sigma_{xy} \end{array}\right\}dz+\sum_{op=1}^{nl_{p}}\int_{-t_{MFCp}/2}^{t_{MFCp}/2}\left\{\begin{array}{c} \sigma_{xx}\\ \sigma_{yy}\\ \sigma_{xy} \end{array}\right\}dz+\sum_{ok=1}^{nl_{k}}\int_{-t_{MFCk}/2}^{t_{MFCk}/2}\left\{\begin{array}{c} \sigma_{xx}\\ \sigma_{yy}\\ \sigma_{xy} \end{array}\right\}dz,$$
(4)$$\left\{\begin{array}{c} M_{x}\\ M_{y}\\ M_{xy} \end{array}\right\}=\int_{-t_{MFCa}/2}^{t_{MFCa}/2}z\left\{\begin{array}{c} \sigma_{xx}\\ \sigma_{yy}\\ \sigma_{xy} \end{array}\right\}dz+\sum_{op=1}^{nl_{p}}\int_{-t_{MFCp}/2}^{t_{MFCp}/2}z\left\{\begin{array}{c} \sigma_{xx}\\ \sigma_{yy}\\ \sigma_{xy} \end{array}\right\}dz+\sum_{ok=1}^{nl_{k}}\int_{-t_{MFCk}/2}^{t_{MFCk}/2}z\left\{\begin{array}{c} \sigma_{xx}\\ \sigma_{yy}\\ \sigma_{xy} \end{array}\right\}dz,$$
(5)$$\left\{\begin{array}{c} Q_{x}\\ Q_{y} \end{array}\right\}=\int_{-t_{MFCa}/2}^{t_{MFCa}/2}\frac{5}{6}\left(1-\frac{4s_{3}{}^{2}}{t_{MFCa}{}^{2}}\right)\left\{\begin{array}{c} \tau_{yz}\\ \tau_{xz} \end{array}\right\}dz+\sum_{op=1}^{nl_{p}}\int_{-t_{MFCp}/2}^{t_{MFCp}/2}\frac{5}{6}\left(1-\frac{4s_{3}{}^{2}}{t_{total\_MFCp}{}^{2}}\right)\left\{\begin{array}{c} \tau_{yz}\\ \tau_{xz} \end{array}\right\}dz+\sum_{ok=1}^{nl_{k}}\int_{-t_{MFCk}/2}^{t_{MFCk}/2}\frac{5}{6}\left(1-\frac{4s_{3}{}^{2}}{t_{total\_MFCk}{}^{2}}\right)\left\{\begin{array}{c} \tau_{yz}\\ \tau_{xz} \end{array}\right\}dz,$$
(6)$$\left\{\begin{array}{c} T_{rx}\\ T_{ry} \end{array}\right\}=\int_{-t_{MFCa}/2}^{t_{MFCa}/2}\frac{5}{6}z\left(1-\frac{4s_{3}{}^{2}}{t_{MFCa}{}^{2}}\right)\left\{\begin{array}{c} \tau_{yz}\\ \tau_{xz} \end{array}\right\}dz+\sum_{op=1}^{nl_{p}}\int_{-t_{MFCp}/2}^{t_{MFCp}/2}\frac{5}{6}z\left(1-\frac{4s_{3}{}^{2}}{t_{total\_MFCp}{}^{2}}\right)\left\{\begin{array}{c} \tau_{yz}\\ \tau_{xz} \end{array}\right\}dz+\sum_{ok=1}^{nl_{k}}\int_{-t_{MFCk}/2}^{t_{MFCk}/2}\frac{5}{6}z\left(1-\frac{4s_{3}{}^{2}}{t_{total\_MFCk}{}^{2}}\right)\left\{\begin{array}{c} \tau_{yz}\\ \tau_{xz} \end{array}\right\}dz,$$
where:

*nl_p_*—the amount of the electrode layers of the MFC element, *op = 1..nl_p_*;

*nl_k_*—the amount of the Kapton layers of the MFC element, *ok = 1..nl_k_*;

*nl*—the total amount of layers of the MFC element: *nl = nl_p_ + nl_k_ +* 1;

*t_MFCa_*—the thickness of the active layer of the MFC element; 

*t_MFCp_*—the thickness of the passive layer of the MFC element;

*t_MFCk_*—the thickness of the Kaption layer of the MFC element;

*t_total_MFCp_*—the total thickness of the passive layers of the MFC element;

*t_total_MFCk_*—the total thickness of the Kaption layers of the MFC element.

In the case of determining the electrical behavior of this laminate piezo structure, the layer-wise approach has been used. Then, according to this method, electrical displacement of this piezo-composite for the active piezoelectric layer is expressed in the following form:(7)$$D_{3}=\int_{-t_{MFCa}/2}^{-t_{MFCa}/2}\left\{E_{3}\right\}dz,$$
where:

$D_{3}$—the electrical displacement of the active layer of the MFC element

$E_{3}$—the electrical field of the MFC element with the vertical polarization.

Determining the representative mechanical behavior of the smart structure with the attached laminate to its surface is also required and is relevant for SHM. To do this, the degenerated shell theory with an implicit curvature [38] is used. Displacements, strains, and the electrical field can be written then as a function of the nodal degree of freedom of the finite element in the following form (Equations (8)–(10)), respectively:(8)$$u_{i}=u_{i}^{n}\phi_{n},\;\theta_{k}=\theta_{k}^{n}\phi_{n},\;\varphi_{p}=\varphi_{p}^{n}\phi_{n},$$
(9)$$\left\{\begin{array}{c} \varepsilon_{11}\\ \varepsilon_{22}\\ \varepsilon_{12}\\ \gamma_{23}\\ \gamma_{13} \end{array}\right\}=\left(\begin{array}{c} B_{m}+B_{b0}+s_{3}B_{b1}\\ B_{s}+B_{t0}+s_{3}B_{t1} \end{array}\right)\tilde{u},$$
(10)$$E_{3}=B_{\phi }\tilde{\varphi },$$
where: $\phi_{n}$ is the shape function for n-th node of the finite element, $B_{m}$, $B_{b0}$, $B_{b1}$, $B_{s}$, $B_{t0}$, $B_{t1}$ are the curvature-displacement components calculated versus in-plane membrane strains (*m*), *b*_0_ is the uniform term of in-plane bending strains, *b*_1_ is the linear term of in-plane bending strains, *s* is the out-of-plane shearing distortions, *t*_0_ is a uniform term of out-of-plane torsions, and *t*_1_ is a linear term of out-of-plane torsions, respectively.

Taking in Equations (8)–(10) to account, the strains, displacements, and electrical potentials of the laminated elements can be expressed in terms of the nodal variables. Subsequently, taking a solution of the elemental equilibrium equation adopted from [38] into account, equations for the piezoelectric problem of laminate structure where *w_i_* and *w_j_* are Gauss’ Quadrature weights, can be expressed as
(11)$$\begin{array}{l} M\ddot{u}+C\dot{u}+K_{uu}u+K_{u\varphi }\varphi =F\\ K_{\varphi u}u+K_{\varphi \varphi }\varphi =Q \end{array}$$
where:(12)$$M=\sum_{i,j=1}^{j=3}\det(J^{-1})w_{i}w_{j}\rho \left[\frac{h}{2}H_{0}^{T}H_{0}+\frac{h^{2}}{4}\left(H_{1}^{T}H_{0}+H_{0}^{T}H_{1}\right)+\frac{h^{3}}{12}H_{1}^{T}H_{1}\right],$$
(13)$$K_{uu}=\sum_{i,j=1}^{j=3}\det(J^{-1})w_{i}w_{j}\left({\left[B_{u}^{m}\right]}^{T}\left[\begin{array}{cc} A & B\\ B & D \end{array}\right]\left[B_{u}^{m}\right]\right)+\sum_{i,j=1}^{j=2}\det(J^{-1})w_{i}w_{j}\left({\left[B_{u}^{t}\right]}^{T}\left[\begin{array}{cc} G & G_{h}\\ G_{h} & H \end{array}\right]\left[B_{u}^{t}\right]\right),$$
(14)$$K_{\varphi u}=\sum_{i,j=1}^{j=3}\det(J^{-1})w_{i}w_{j}\left({\left[B_{ut}\right]}^{T}\left[\begin{array}{c} {\bar{e}}_{1}\\ {\bar{e}}_{2} \end{array}\right]\left[B_{ut}\right]\right),$$
(15)$${Κ}_{\varphi \varphi }=\sum_{i,j=1}^{j=3}\det(J^{-1})w_{i}w_{j}B_{u\varphi }{}^{T}\bar{d}B_{u\varphi },$$
(16)$$C=\alpha M+\beta K_{uu},$$
(17)$$B_{u}^{m}=\left[\begin{array}{c} B_{b0}^{}+B_{b0}^{}\\ B_{b1}^{} \end{array}\right],\;B_{u}^{t}=\left[\begin{array}{c} B_{t0}^{}+B_{t0}^{}\\ B_{t1}^{} \end{array}\right],\;B_{u\varphi }^{}=\left[\begin{array}{c} B_{b0}^{}\\ B_{b1}^{} \end{array}\right],\;H_{0_{i,5(n+1)+j}}=\delta_{ij}\phi_{n},H_{1_{i,5(n+1)+j}}=\delta_{\left(i,j+3\right)}\frac{h}{2}\phi_{n}H_{ij}^{n}.$$

From a numerical point of view, the electrical and mechanical degree of freedom, as well as the generalized mass, damping and stiffness matrices need to be transformed to modal coordinates in such a way that the nodal variables for a given element can be obtained in a single vector ordered by node numbering. In order to do this, the transforming matrix R should be used. Then, a general mechanical-electrical system in modal coordinates can be express in the following form, respectively:(18)$$\hat{u}={\left\{\begin{array}{ccccccccc} u_{1}^{1} & u_{2}^{1} & u_{3}^{1} & \theta_{x}^{1} & \theta_{y}^{1} & \varphi_{p(1)}^{1} & \ldots & \varphi_{p(last)}^{1} & \ldots \\ u_{1}^{n} & u_{2}^{n} & u_{3}^{n} & \theta_{x}^{n} & \theta_{y}^{n} & \varphi_{p(1)}^{n} & \ldots & \ldots & \varphi_{p(last)}^{n} \end{array}\right\}}^{T},$$
(19)$$\hat{u}=R\left\{\begin{array}{c} \tilde{u}\\ \tilde{\varphi } \end{array}\right\},$$
(20)$$\hat{K}=R\left[\begin{array}{cc} K_{uu} & K_{u\varphi }\\ K_{\varphi u} & K_{\varphi \varphi } \end{array}\right]R^{T},$$
(21)$$\hat{M}=R\left[\begin{array}{cc} M & 0\\ 0 & 0 \end{array}\right]R^{T},$$
(22)$$\hat{C}=R\left[\begin{array}{cc} C & 0\\ 0 & 0 \end{array}\right]R^{T},$$
(23)$$\hat{F}=R\left\{\begin{array}{c} F\\ Q \end{array}\right\},$$
(24)$$\hat{M}\ddot{\hat{u}}+\hat{C}\dot{\hat{u}}+\hat{K}\hat{u}=\hat{F}$$

## 3. Numerical Analysis of a Smart Beam as a Laminated Structure

The numerical calculations of the cantilever beam of a length of 380 mm, width of 31 mm, and a thickness of 1.8 mm, with two piezo-stripe elements (MFC 8528 P2), consisting of a three-dimensional homogenized material in the active layer, is described in this section. The parameters of a homogenized material for the MFC taken from [40] are collected in Table 1. To assess the values of damage identification metrics, the considered structures (intact and damage structures with first one, two and three drilled holes) were modelled in the Ansys software with the assumption that the first MFC element is located in the distance of 40 mm from the fixed end of the beam, while the second one is at a distance of 15 mm from the free end of the beam. Taking into account the structure of the cantilever beam with MFC attached to its top surface, the host beam structure is modelled by using an 8-node coupled-brick element Solid186. For the MFC element, the electrode and Kapton layers are modelled similarly with the use of a Solid186 element, while the piezoelectric fiber layer with a homogenized material is modelled by using a Solid226-node coupled brick element. In addition, it has been assumed that modelling of the adhesive layer can be omitted due to its thickness which is less than 15 µm. Finally, this leads to determining the computational model of the considered structure shown in Figure 2 consists of 120 of finite elements Solid226 of the length of 10 mm for the cantilever beam, 102 elements of type Solid226 for each passive layer (electrode and Kapton) and 102 elements of type Solid226 for the active layer.

In the first step of numerical calculations, the eigenvalue problem of such a modelled structure is solved by using the Ansys software. For this purpose, the behavior of three different structures (one intact and two damage structures with one hole and two holes, respectively, of the diameter of 8 mm) are compared in the selected frequency range 1–400 Hz. The example of the damage structure with two holes is shown in Figure 3. The obtained eigenvalues for each structure are listed in Table 2. 

Taking into account the obtained result shown in Table 2, it can be noticed that the values of the first four lowest natural frequencies of the damage structures for the increasing number of holes in the structure decreases slightly. The obtained effect is insignificant from the point of view of considering an SHM system since the decrease is of about 0.5% versus the values of eigenvalues of the intact structure. This is thus can be omitted in further analysis. 

Next, a harmonic analysis for each structure is performed. To do this, the computational models of intact and damage structures with shell models of the MFC elements were excited by an impulse load of 48N magnitude, while the vertical displacement was taken from specific nodes of brick models of piezo composites MFC1 and MFC2, respectively. Taking into account the results presented in Figure 4, it can be observed that the accuracy between the frequency responses of intact and damage structures is high, especially in resonance regions, where the vibration amplitudes of the damage structure are lower than that of the intact structure (see Figure 5). This leads to difficulties related to proper identification and interpretation of damage in the structure and the calculation of damage identification metrics as damage indicators. This issue is taken up in detail in the next section. 

Further analysis of the harmonic responses performed for the FEM models of the intact and damage structures indicates that the distance between the location of the measurement and excitation is the cause of obtaining two different kinds of systems from a control strategy view point, linked to collocated for MFC1 sensor and non-collocated for MFC2 sensor, respectively. An example of the frequency response for the collocated system is the upper diagram shown in Figure 4, where resonance and anti-resonance frequencies are occurring. An example of the frequency response of the non-collocated system is presented in the lower diagram in Figure 4 where the increase of the distance between the point of measurement related to the location of the piezo MFC2 and the point of impact lead to omitting the first antiresonance frequency. Consequently, a proper location of the piezo-sensor on the structure should be also considered for SHM deployment.

## 4. Experimental Verification

The process of frequency response function (FRF) verification of the intact and damage structures are further used to compute damage identification metrics. Two piezo-patch sensors, MFC 8528 P2, were attached to the host aluminium beam structure with the help of an adhesive UHU Plus (Figure 6). The first piezo MFC1 is located 40 mm from the fixed end, while the second one in the distance of 15 mm from the free end of the beam. The laboratory stand is retrofitted into the modal hammer developed by Bruel and Kjaer used to excite the considered structures to vibrations. The data acquisition module PXI 4496 is used to measure and record vibrations from the beam. 

Taking numerical investigations of Section 3 into account, experiments on a real structure were carried out. An impulse load with a magnitude of 48 N is applied to the structure to a chosen point located 10 mm before the piezo MFC1 and 30 mm from the fixed end of the beam while structural vibrations are measured with both laminate composites, MFC1 and MFC2, respectively.

First, the intact structure is investigated by applying the aforementioned impulse load to the beam at 4.2 s of measurement. Taking into account the recorded voltage from both piezo elements shown in Figure 7, it can be seen that the transient response measured with MFC2 is longer than that measured with MFC1. This is due to the fact that the piezo MFC1 is located closer to the fixed end of the beam where the damping is higher. Further analysis of the intact structure behavior requires transformation of input/output signals to the frequency domain and determining two frequency responses functions (FRFs) at MFC1 and MFC2 locations, respectively, as presented in Figure 8. 

In Figure 8, collocated piezo MFC1 and non-collocated piezo MFC2 aspects are observed and this allows to verify FRFs determined from computational models. The frequency response for piezo MFC1 has interchangeable character of the resonance and anti-resonance frequencies, while in the case of MFC2, the generated frequency response is without the first anti-resonance frequency. Thus, the distance between the sensor and the actuator is crucial to describe the behaviour of the structure. 

Further analysis of the recorded frequency responses from both piezo-sensors showed also a high convergence between them, especially in the resonance areas, where the amplitude of vibrations from tests is close to the amplitude from the numerical model. In other areas it can observe mismatch between both FRFs that are due to a heterogenous adhesion between the piezo elements and the host structure and nonlinearity of the MFC, especially in the strain-displacement relation. In the effect, the real frequency response generated on the basis of the noisy measurement signal contains additional slight amplitude peaks especially in higher frequencies. However, despite these discrepancies, from monitoring point of view, it can be still conclude that those responses properly verify the numerical responses. 

In the next step, an experimental test was carried out for the damage structure with one hole drilled in the distance of 25 mm from the end of the MFC1 piezo-patch sensor and 150 mm from the place of impact. Similarly to the previous case, this structure was excited to vibration again by using a modal hammer and applying the impulse load with the magnitude of 48 N at the same point. As a result, two measurement signals from both piezo-composites were measured by a PXI module that next allow to generate two separate frequency response functions showed in Figure 9.

Observing diagrams in Figure 9, it can be noticed again that the experimental frequency response functions are close to the frequency responses obtained based on the numerical model, especially in the resonance areas of the first four natural frequencies. In this case, it can be seen that the natural frequencies of the structure measured with the help of both piezo-composites, MFC1 and MFC2, are identical with those calculated from the numerical model, while their amplitudes, especially for those measured by using the MFC1, are less convergence. This behaviour results from higher damping in the real structure than it was assumed for the numerical model. Moreover, as it was just mentioned, it is caused by heterogenous adhesion between the sensor and the host structure, nonlinearity of the piezo sensors, as well as the noisy measurement signal that is used to generate real frequency response. The mismatch is also representative of typical tests.

In the same way, the structure with two holes located very close to the MFC1 sensor was investigated. In this case, the second hole was located 12 mm from the first one and 37 mm from the end of the piezo MFC1. Taking into account Figure 10, it can be observed that the amplitudes of the structure vibration on the generated FRF from the piezo MFC1 are close to the amplitudes vibrations calculated based on the numerical model. Another behaviour that can be seen in the case of the FRF analysis from the piezo MFC2 is the high convergence only in resonant areas. The main reasons of this mismatch can be attributed to a heterogenous adhesion between the bottom surface of the piezo MFC2 and the top surface of the aluminium beam, nonlinearity of the piezo-composite and noisy measurement signal. Again, despite some discrepancies between them located outside the resonance areas, the FRF generated from the lab stand can be assumed as correct. 

The last step of the experimental test was collecting all the generated FRF from both piezo sensors—MFC1 and MFC2—to perform their analysis and assess the real value of the damage identification metrics. Taking into account Figure 11, it can be observed that the increasing number of holes in the damage structure and the decreasing stiffness of the structure in chosen areas of the beam do not lead to a change in the natural frequencies but affects only the amplitude of the structure in the resonance areas. It this way, the conclusion from the analysis of the computational model has been verified. Further analysis of these diagrams indicates also that the decrease of the beam stiffness resulting from drilling the holes in the areas located very close the MFC1 sensor leads also to the amplitude increase of vibrations measured by the MFC1 sensor but only for the first lowest natural frequencies. In the case of the piezo MFC2, it can be observed that drilling one hole leads firstly to the increase of the vibration amplitude while drilling another hole leads to its decrease. A similar effect is also presented in Figure 12 where the power spectrum of measurement signals from both piezo-sensors is analyzed. Finally, taking into account the generated FRF’s and the power spectrum of signals from the piezo composites, it can be concluded that those diagrams are insufficient to identify the damage in the structure properly. For this reason, the damage identification metrics should be calculated. 

## 5. Damage Identification Metrics and Discussion

The numerical and experimental investigations of the intact structure and damage structures presented in the previous sections showed problems with a proper identification of damage in the structure only in terms of FRFs, because the dynamics of these structures is scattered when the frequency increases. Therefore, in order to assess the precision of the damage type and the damage localization, the damage identification metrics should be calculated. Taking this fact into account, five different damage identification metrics M1–M5 taken from [41,42,43,44,45] were calculated for each considered damage structures, and the results of the calculations are presented in this section. In addition, in order to perform better analysis, the metric M2 for the intact structure is calculated as a reference value as well as the damage indicators M1–M5 on the basis of the computational model with three holes. The calculation of damage indices M1, M3, M4, and M5 cannot be done for a healthy baseline because they represent a relationship between the damaged and the intact structure, and consequently a reference M2 marker can link the performances together. 

In the first step, their values were calculated based on the frequency responses from the numerical models and then from the FRF generated (Figure 8, Figure 9 and Figure 10) from the laboratory setup. 

The metrices considered in this paper can be divided into two groups: quantitative indicators M1–M3 given by Equations (25)–(27)and qualitative indicators M4–M5 given by Equations (28) and (29). The calculation of these metrics for the first group were performed in terms of a specific frequency value to assess the damage localization in the structure, while in the case of the second group, in terms of the selected frequency range 1–400 Hz, to assess the level of damage. Finally, the results obtained from the computational model considering the damage structure with three holes were collected in Table 3, while results for the experimental response without the structure with the most number of holes are presented in in Table 4. In addition, in order to easier analyze, the damage metric M2 for the intact and damage structures with one and two holes is also presented in the form of a diagram in Figure 13.
(25)$$M_{1}=\max\left({\left.\frac{H_{I}(f_{i})}{H_{d}(f_{i})}\right|}_{P1},{\left.\frac{H_{I}(f_{i})}{H_{d}(f_{i})}\right|}_{P2}\right),$$
where:

$H_{I}(f)$, $H_{d}(f)$ denotes frequency response of the intact and damage structure, respectively.
(26)$$M_{2}=\max\left(\frac{{\left.H_{d}(f_{i})\right|}_{P2}}{{\left.H_{d}(f_{i})\right|}_{P1}}\right),$$
(27)$$M_{3}=\frac{\log(H_{D}(f_{i})-H_{I}(f_{i}))}{\log(H_{I}(f_{i}))}*100\%.$$

**Figure 13 sensors-21-06796-f013:**
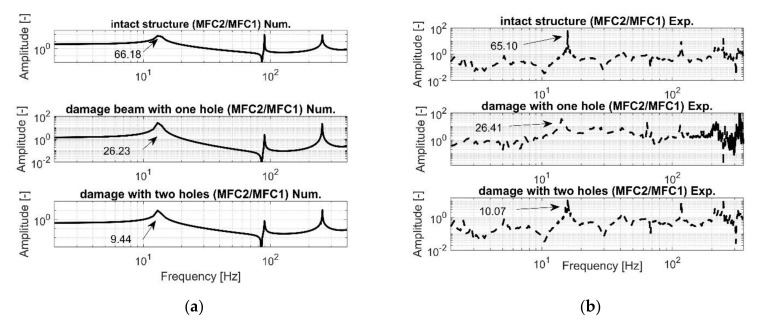
The comparison of damage identification metric M2 of the damage structures with one hole and two holes calculated based on (**a**) numerical approach, (**b**) experimental approach.

**Table 3 sensors-21-06796-t003:** Results of the damage metrics M1, M2, and M3 calculated in terms of numerical models.

Frequency Response Function	Damage Metrics				
	Metric M1		Metric M2	Metric M3 [%]	
Frequency [Hz]	10.1	60.8	13.5	9.97	60.8
Sensor	MFC_1	MFC_2	-	MFC_1	MFC_2
Intact structure	-	-	**66.18**	-	-
Damage structure (1 hole)	5.874	1.340	26.23	234.1	203.2
Damage structure (2 holes)	7.403	1.682	9.44	253.0	237.5
Damage structures (3 holes)	9.508	1.956	3.37	286.3	265.3

**Table 4 sensors-21-06796-t004:** Results of the damage metrics M1, M2, and M3 calculated in terms of the generated FRF from the lab stand.

Frequency Response Function	Damage Metrics				
	Metric M1		Metric M2	Metric M3 [%]	
Frequency [Hz]	10.1	60.8	13.9	10.1	60.5
Sensor	MFC_1	MFC_2	-	MFC_1	MFC_2
Intact structure	-	-	**65.10**	-	-
Damage structure (1 hole)	5.399	1.278	26.41	230.1	209.1
Damage structure (2 holes)	7.620	1.569	10.07	257.0	241.5

Taking into account the results collected in Table 3 and Table 4 and Figure 13, it can be noticed that the experimental tests and the obtained values of the damage metrics M1, M2, and M3 verify in their numerical results. The analysis of the values collected in Table 3 and Table 4 show that a gradual decrease of the structure stiffness in a chosen area of the structure leads to the increase of the values of M1 and M3. The inverse effect can be obtained in the case of the analysis of the damage metric M2 (Figure 13), where the increasing number of holes in the structure relative to the intact structure leads to a decrease of its maximum value. It is, however, important to note that all metrics exhibit a monotonicity of calibration against damage, which is important for SHM. Further analysis of these tables also shows that the calculated maximum values of the damage metrics M1 and M3 for the piezo-composite for MFC1 cases are higher than their values calculated for the piezo-patch MFC2. This leads to a conclusion that the damage in the structure is located closer the piezo MFC1 and the piezo MFC2. With adequate sensors, this can lead to the localization of damages better. The actual demand of the spacing of sensors will eventually depend on the demands of detection of the feature of interest in terms of extent and resolution, noise in the measured signal, and the excitation that generates the responses.

Next, the values of the qualitative indicators M4 and M5, given in Equations (28) and (29), were calculated to assess the level of damage in the structure in the selected frequency range of 1–400 Hz, with a frequency increment *Δf* of 0.00024 Hz (reciprocal of sampling frequency *f_s_* = 4096 Hz). Similar to previous cases, the first frequency responses functions from the numerical model (Figure 7, Figure 8 and Figure 9) were taken to calculate these values, and next, the FRF from the laboratory experiment to verify them. The obtained results for the damage metric M4 are shown in Figure 14, while the results for the damage metric M5 in Figure 15.
(28)$$M_{4}=\frac{\Delta f}{f_{high}-f_{low}}\sum_{i=1}^{n}\left|\frac{H_{d}(f_{i})-H_{I}(f_{i})}{H_{I}(f_{i})}\right|,$$
where:

*Δf*—the frequency increment; 

*f_high_*—the upper frequency;

*f_low_*—the lower frequency.
(29)$$M_{5}=\frac{\sum_{i=1}^{n}\left|H_{d}(f_{i})-H_{I}(f_{i})\right|}{\sum_{i=1}^{n}\left|H_{I}(f_{i})\right|}.$$

**Figure 14 sensors-21-06796-f014:**
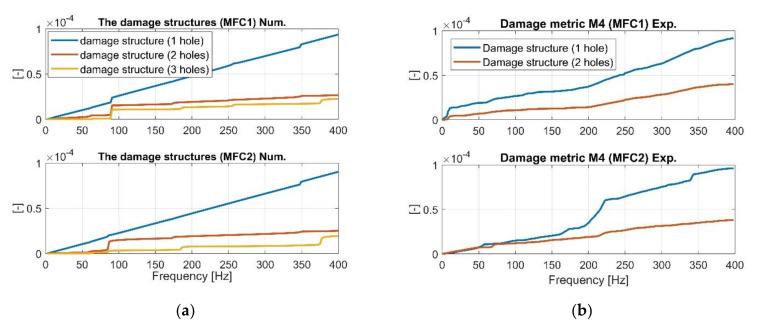
The comparison of damage identification metric M4 of the damage structures with one hole and two holes calculated based on (**a**) numerical approach, (**b**) experimental approach.

**Figure 15 sensors-21-06796-f015:**
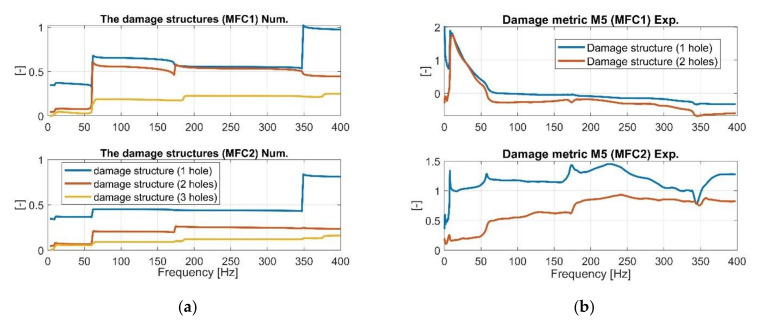
The comparison of damage identification metric M5 of the damage structures with one hole and two holes calculated based on (**a**) numerical approach, (**b**) experimental approach.

As observed in Figure 14b and Figure 15b, the experimental results of the damage metrics M4 and M5 carried out for only two damage structures with one and two holes allow verifying the values obtained from the computational model. Analysis of M4 indicates a decrease of its values for a gradually increasing number of the drilled holes in a chosen area of the beam even for more damaged structure (see orange line in Figure 14a). A similar observation was noted during the analysis of M5 where its value for the structure with three holes were the lowest. Overall, damage metrics M1–M5 are useful to identify damage and its localization, and can support SHM for beam-like structures. Moreover, these results can be useful to build equivalent damage model and also create a fundamental, low-fidelity system which can lend itself to further studies. 

## 6. Summary and Conclusions

The use of piezoelectric patches in SHM has expanded the possibilities of use of energy harvesting in recent times. Nonlinearity in the piezo patches with a potential application for SHM has led to investigations in this paper on structures composed of thin piezo-stripes by creating computational models for them. With the current focus on using traditional and modern sensors to aid digital twinning and model updating, such a focus on the behavior of the sensors becomes even more important. Composite structures are making new inroads into a range of sectors, including renewable energy, and so this example is also relevant for future expansion in terms of sustainability of such solutions. 

Taking this into account, the stress–strain effect in the laminate structure was analyzed first in this paper to create a fundamental background model. Next, a modal analysis of the chosen structures (intact and damaged) with piezo patch MFCs were carried out using FEM, establishing a homogenized model of MFC elements. Results presented in Figure 4 and Figure 5 indicate that a gradual increase of the number of the holes drilled in the beam in a chosen area slightly affects the values of the resonance and antiresonance frequencies and leads to a decrease of the vibration amplitude due to changes in local stiffness. Further analysis of response to harmonic loading, performed on the FEM models for the intact and damage structures indicates that the distance between the measurement location and excitation leads to collocated (sensor MFC1) and non-collocated (sensor MFC2) systems. Taking the FRF of the collocated system into account (Figure 8a, Figure 9a and Figure 10a) the inversion of resonance and anti-resonance frequencies is observed. For non-collocated system (Figure 8b, Figure 9b and Figure 10b), the increase of distance between the sensor and the location of excitation leads to the estimation of FRFs while omitting the first antiresonance frequency. 

Next, FRFs of the FEM models of the intact and damaged structures are used to assess the damage location. Five different damage indexes were calculated in this regard, comprising of three quantitative and two qualitative indicators, estimated as a function of locations of the sensors on the structure. Estimated damage indices in Table 3 show that a gradual decrease of the beam stiffness leads to consistent and monotonic change of the indexes (26% increase of M1, 20–30% increase of M3 and decrease of M2) with respect to the reference value. This consistency of the damage indicators in the presence of realistic conditions is desirable and is indicative of their robustness.

Additionally, taking into account the values in Table 3 and Table 4, it can be noticed that the damage of the structure is located closer to the MFC1 sensor and consequently, higher values of damage indicators closer to the harvesting sensors can also be used to identify the approximate location of the damages. With reasonably spaced sensors in the context of damage detection resolution requirements, this can provide information in terms of detection of the damage location. The qualitative metrics also show a decrease in values for increasing number of holes in the structure, as observed from numerical simulations. This consistency of multiple metrics to describe same damage changes also opens up the possibility of using combined metrics to have a more robust detection scheme.

Experimental investigations carried out in the laboratory (one intact and two damage) with an attached PEH allowed to verify the numerical results indicate that estimated FRFs from piezo-sensors MFC1 and MFC2 are consistent with changes due to damage. Subsequent analysis (Figure 11 and Figure 12) also confirms that the beam stiffness decreasing in a chosen region by drilling an increasing number of holes slightly affects the values of resonances and antiresonances, but significantly affects their amplitude within the range of low frequencies. This is especially illustrated in the vicinity of the first natural frequency where drilling subsequent holes leads to the increase of their values (Figure 12a—MFC1), while in the case of the measurement using MFC2 sensor, the amplitude of vibrations increases and then decreases. Heterogenous adhesion between the harvesting elements and the host structure can lead to such a situation. Summarizing the experimental tests, it can be said that the damage location on the basis only on the analysis of FRF is difficult for energy harvesting and it must be processed further to create relevant markers of damage detection. The proposed damage metrics in this paper illustrate how such markers can be developed and combined, especially in an output-only context. The results of Table 4 indicate similar trends of the proposed metrics as compared to what was observed through numerical simulations. Higher values of damage metrics were observed for sensor closer to the damage (Figure 14b and Figure 15b) along with distinctive and consistent difference over the entire testing range of 1–400 Hz.

This work can act as a reference point for modelling and the expectation of performance of such energy harvesting based SHM sensors for applications in civil/mechanical systems.

## Figures and Tables

**Figure 2 sensors-21-06796-f002:**
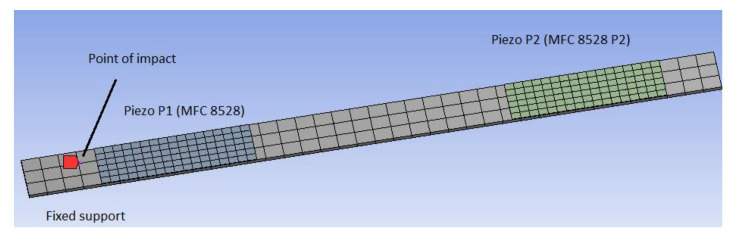
The numerical model of the smart intact structure with both piezo composites attached to the host structure.

**Figure 3 sensors-21-06796-f003:**
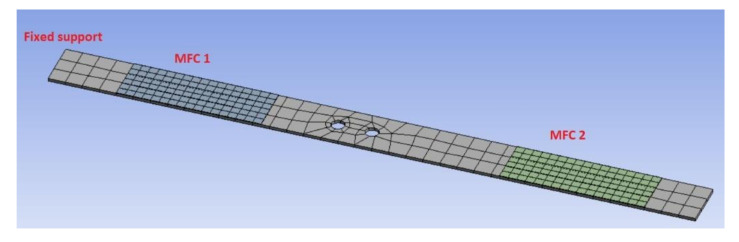
The numerical model of the damage structure with two holes drilling in region close to the end of the piezo-patch MFC1.

**Figure 4 sensors-21-06796-f004:**
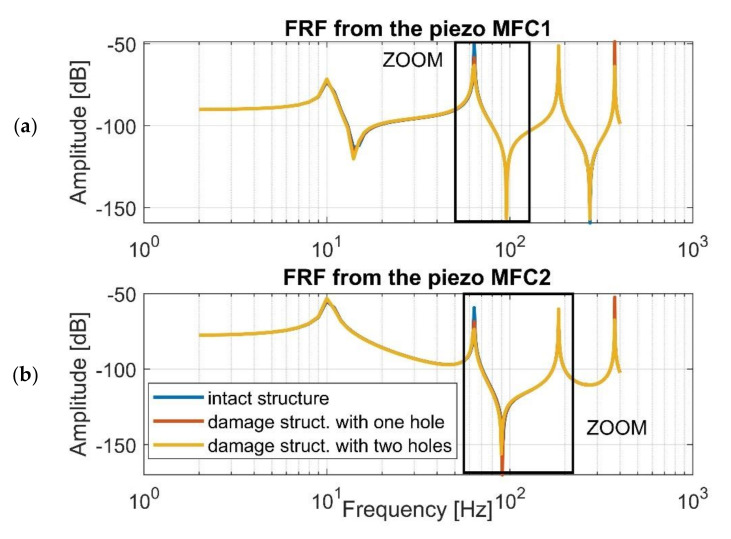
The frequency response function for the intact and damaged structures calculated in the selected frequency range of 1–400 Hz in terms of (**a**) the piezo MFC1 location, (**b**) the piezo MFC2 location.

**Figure 5 sensors-21-06796-f005:**
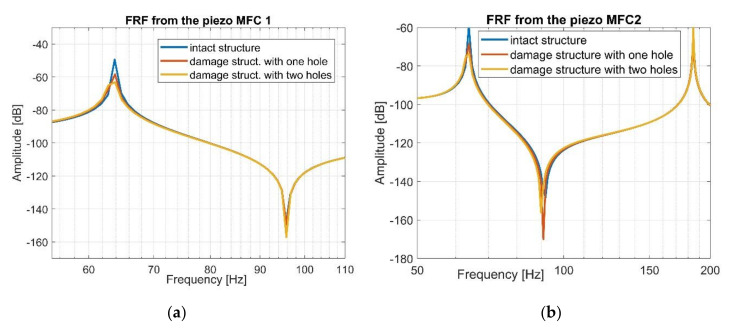
The frequency response function for the intact and damaged structures calculated in the selected frequency range in terms of (**a**) the piezo MFC1 location, (**b**) the piezo MFC2 location.

**Figure 6 sensors-21-06796-f006:**
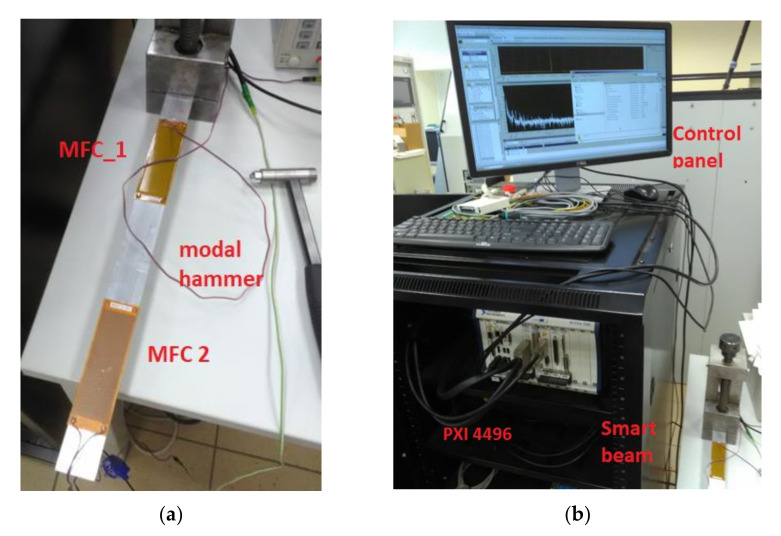
The view of a real laboratory stand during a lab test (**a**) the smart beam with both piezo-composites MFC 8528 P2, (**b**) the data acquisition module PXI 4496a.

**Figure 7 sensors-21-06796-f007:**
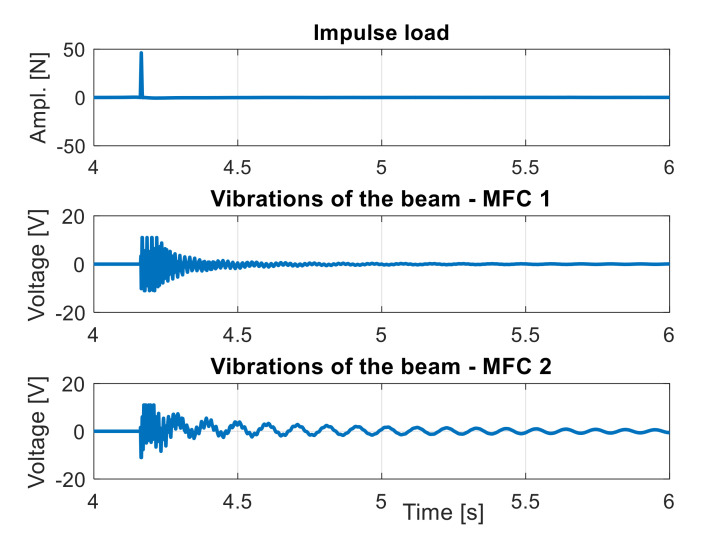
The excitation signal and measurement signals measured by MFC elements during analysis of the intact structure.

**Figure 8 sensors-21-06796-f008:**
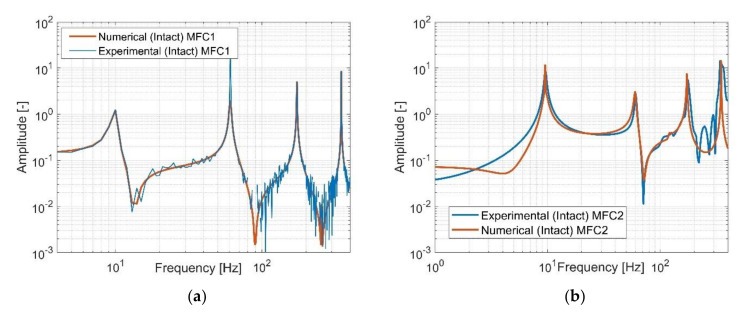
The comparison of the amplitude plots of the intact structure measured by using (**a**) piezo-patch MFC1, (**b**) piezo-patch MFC2.

**Figure 9 sensors-21-06796-f009:**
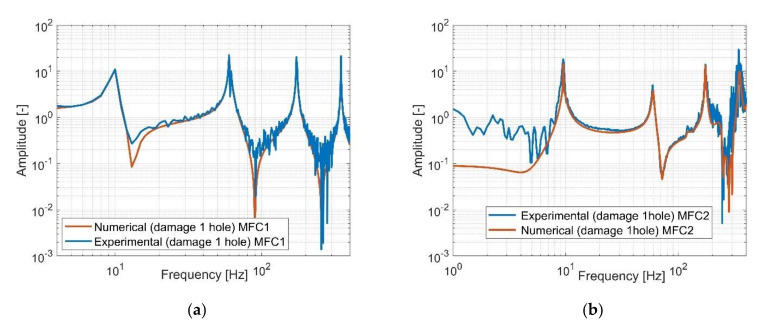
The comparison of the amplitude diagram of the damage structure with one hole measured by using (**a**) piezo-patch MFC1, (**b**) piezo-patch MFC2.

**Figure 10 sensors-21-06796-f010:**
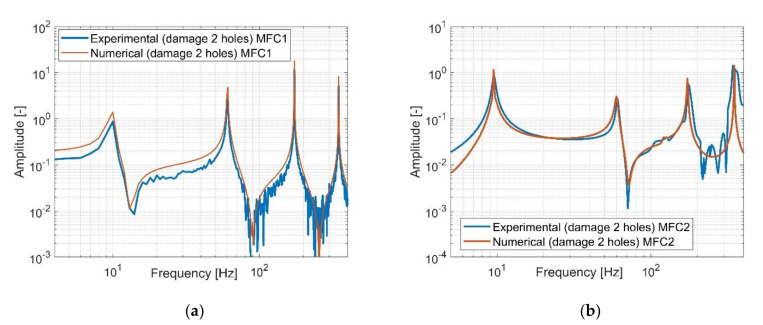
The comparison of the amplitude plots of the damage structure with two drilled holes measured by using (**a**) piezo-patch MFC1, (**b**) piezo-patch MFC2.

**Figure 11 sensors-21-06796-f011:**
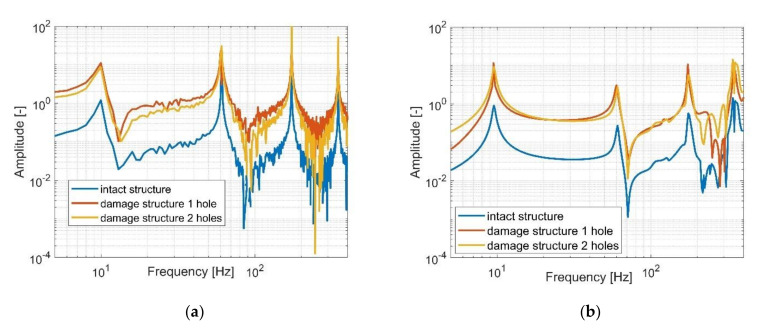
The comparison of the FRF of the intact beam and damage structures with one hole and two holes measured by (**a**) piezo-patch MFC1, (**b**) the piezo-patch MFC2.

**Figure 12 sensors-21-06796-f012:**
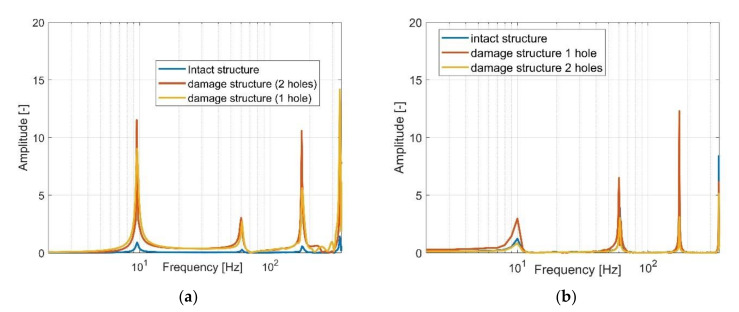
The comparison of power spectrum of the intact beam and damage structures with one hole and two holes measured by (**a**) piezo-patch MFC1, (**b**) the piezo-patch MFC2.

**Table 1 sensors-21-06796-t001:** Material properties of homogenized MFC layer of MFC8528 P2.

Mechanical Parameters						
Young’s Modulus (GPa)	Poisson’s Ratio (-)	Shear Modulus (GPa)	Piezoelectric Charge Coefficient (pC/N)	Relative Permittivity (-)		
E_x_ 31.6	v_xy_ 0.4	G_xy_ 4.9	d_31_ −173	ε_r_^T^ 2253		
E_y_ 17.1	v_yz_ 0.2	G_yz_ 2.5	d_32_ −150			
E_z_ 9.5	v_xz_ 0.4	G_xz_ 2.4	d_33_ 325			
Geometrical parameters						
**overall length [mm]**	**overall width** **[mm]**	**active length** **[mm]**	**active width [mm]**	**thickness of fiber layer [µm]**	**thickness of electrode layer [µm]**	**thickness of Kaption layer [µm]**
103	31	85	28	180	25	30

**Table 2 sensors-21-06796-t002:** Values of natural frequencies of the intact structure and damage structures.

Mode of Vibration	Eigenvalues [Hz]		
	Intact Structure	Damage with One Hole	Damage with Two Holes
First	10.27	10.23	10.20
Second	63.74	63.52	63.25
Third	183.04	182.84	182.85
Fourth	369.79	369.12	368.13

## Data Availability

Data for the experiments are available from the authors on request.

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
