# Peer review of "Numerical Analysis and Experimental Verification of Damage Identification Metrics for Smart Beam with MFC Elements to Support Structural Health Monitoring"

_sensors, 2021, doi:10.3390/s21206796_

Round 1

Reviewer 1 Report

This paper used macro fiber composite (MFC) to identify the structural healthy monitor of a cantilevered beam.  The manuscript cannot figure out what energy harvester in relation to structural healthy monitor. Even the MFC replaced as piezoelectric film (PVDF), the research can reach out the outcome.

The poor text's structure makes reader misunderstanding the focus of the manuscript. The first paragraph on abstract is completely not necessary. The first four sections can be shorten and re-explanation, because the damage identification is failure by frequency and amplitude of spectrum. But, the inconsistent results in numerical calculation and experimental verification has to be well-explanation on section 4.

Equations (28)-(30) have garbled ASCII code . The most important qualitative indicators in Eqns. (31)-(35) lost the definition on the H(f). The parameter H occurs on Eqns. (9), (10), (16), (17), and (21), but the reviewer doesn't think they are the same mathematical definition. On section 3, the comparison established on intact structure and the beams with one hole and two holes. However, the whole analysis on section 5 is only shown the results in the beams with one hole and two holes, without intact structure. The conclusion is also too long to loss the focus.

Author Response

Comments for Reviewer remarks are included in the attachment.

Reviewer 2 Report

This paper investigated the subject of energy harvesting based structural health monitoring. A numerical computational model is developed with experimental validation. Vibration of three different beam structures are monitored in a controlled manner via two piezoelectric energy harvesters. One is undamaged and the other two is damaged. The comparison and reference are expected to  inform in designing new SHM metrics. However, the reviewer has some comments:

(1) In the paper,  the MFC components is simplified to be uniform material. Is there any difference if a usual piezoelectric element is used? if no difference, maybe MFC should be stressed in the title.

(2) The damage is simulated with holes. Does it represent the general case in real situation? Perhaps cracks are more often. Is there any improvements to be done?

Author Response

Comments for Reviewer remarks are included in the attachment

Reviewer 3 Report

In this paper, the authors established a detailed numerical and experimental reference utilizing fundamentals, commercial software and damages and focused on damage identification in thin structures by using piezo fiber composites. The comparison and reference are expected to be relevant for a wide range of application sectors, useful for the evolving composite industry and also inform in designing new SHM metrics and their performance or assessing established ones in a certain exposure environment combined with specific detection needs. The manuscript was sufficient to be published in the journal Sensors after solving some problem. For example, the paper should be polished carefully cause there are many grammar and symbolic errors such as missing asymbol at the end of paragraph 1 of page 4. And the authors should identify whether is suitable using “Energy Harvesting based Structural Health Monitoring”, because sensors are usually used for structural health monitoring.

Author Response

(The authors gave the same response as above.)

Reviewer 4 Report

The manuscript (sensors-1366983) develops a numerical model for damage identification in MFC piezoelectric elements, modelling a beam structure with two elements operating in the 31 directionality. The model is subsequently verified experimentally using a frequency response function analysis. Finally, the manuscript develops and validates five metrics to investigate the type, extent and location of damage, with the aim of acting as a guide in building future structural health monitoring systems based on the piezoelectric effect.

This manuscript is thorough and builds on previous work in the field to develop a novel methodology. The story is in logical order and the language is clear. However, it is unclear how the metrics (M1 to M5) were developed and why these would be the most appropriate to use going forward. In addition, prior to publication, the following points should be addressed:

  1. All acronyms should be defined (e.g., PVDF on line 75, PZT on line 78). Additionally, it appears in the manuscript that polyvinylidene fluoride and PVDF are differing materials (line 80), whereas PVDF is merely the acronym for poly(vinylidene fluoride).
  2. The piezoelectric constitutive equations present incorrect directionality, which, due to the directions of electromechanical coupling can affect the developed model. The correct form, taking into account the directionality and using the author's notation, is as follows (for reference see the IEEE Standard on Piezoelectricity 176-1987):

    σij=CEijklεkl-ekijEk
    Dk=eiklεkl+dεijEk
  3. What does the e subscript represent in equation 7 for displacement field?
  4. In the paragraph beginning on line 407, the discussion does not correspond to the referenced figure 5. Can this be clarified? Is the figure number incorrect?
  5. In figure 9, a significant amount of space is wasted between 0 s and 4 s, can the active region be enlarged (i.e., show between 4 s and 7 s)?
  6. The manuscript mentions (line 487) some discrepancies occurring outside of the resonance frequency regions in figure 11. Can the author discuss the origin of these discrepancies between the developed model and the experimental data?
  7. While the majority of resonance frequencies do not shift throughout the course of the manuscript, the first anti-resonance frequency in MFC1 appears to shift both with respect to the numerical modelling and to the extent of damage. Can the author provide reasoning for this?
  8. Data in table 3, table 4 and the discussion starting on line 561 for the developed metrics is only presented for the two damaged beams. The metrics data for the undamaged beam should be presented and discussed considering it is the reference sample. Additionally, extrapolating and making conclusions from two data points (for each metric) in the paragraph starting on line 561 is questionable. It would make more sense to do so when the data for the undamaged reference sample is present, to provide at least three data points.

Round 2

Reviewer 1 Report

The revision is well modified following all reviewer's opinions. However, because beam is very fundamental structure in analysis, the experimental results should  almost completely correspond to the numerical calculation. From Figs. 8-10, the difference looks not good enough. The authors must be improvement on the experimental data or modification of numerical model. On the other hand, the reviewer thinks still the last paragraph of the Section I not necessary. And, the conclusion should be illustrated the benefits and disadvantages in damage identification in clear "text" explanation, not in the self-named M1 to M5.

Reviewer 2 Report

The authors had addressed the reveiwer's concerns appropriately. The paper can be accepted now.

Reviewer 4 Report

Upon reviewing the updated version of the manuscript by Koszewnik et al., the manuscript is significantly strengthened by the authors in response to the reviewers' comments. In addition, the response by the authors is clear and addresses my previous concerns. A minor point - in equation 7, the electrical displacement should follow the same directionality as its constituents. In this case, the correct form of the left hand side of the equation is D3. With that in mind, I would recommend this manuscript for publication following minor spelling/grammar corrections.
